# Scalable Laplacian K-modes

**Imtiaz Masud Ziko** *
ÉTS Montreal

**Eric Granger**
ÉTS Montreal

**Ismail Ben Ayed**
ÉTS Montreal

## Abstract

We advocate Laplacian K-modes for joint clustering and density mode finding, and propose a concave-convex relaxation of the problem, which yields a parallel algorithm that scales up to large datasets and high dimensions. We optimize a tight bound (auxiliary function) of our relaxation, which, at each iteration, amounts to computing an independent update for each cluster-assignment variable, with guaranteed convergence. Therefore, our bound optimizer can be trivially distributed for large-scale data sets. Furthermore, we show that the density modes can be obtained as byproducts of the assignment variables via simple maximum-value operations whose additional computational cost is linear in the number of data points. Our formulation does not need storing a full affinity matrix and computing its eigenvalue decomposition, neither does it perform expensive projection steps and Lagrangian-dual inner iterates for the simplex constraints of each point. Furthermore, unlike mean-shift, our density-mode estimation does not require inner-loop gradient-ascent iterates. It has a complexity independent of feature-space dimension, yields modes that are valid data points in the input set and is applicable to discrete domains as well as arbitrary kernels. We report comprehensive experiments over various data sets, which show that our algorithm yields very competitive performances in term of optimization quality (i.e., the value of the discrete-variable objective at convergence) and clustering accuracy.

## 1 Introduction

We advocate Laplacian K-modes for joint clustering and density mode finding, and propose a concave-convex relaxation of the problem, which yields a parallel algorithm that scales up to large data sets and high dimensions. Introduced initially in the work of Wang and Carreira-Perpinán [33], the model solves the following constrained optimization problem for $L$ clusters and data points $\mathbf{X} = \{\mathbf{x}_p \in \mathbb{R}^D, p = 1, \ldots, N\}$:

$$\min_{\mathbf{Z}} \quad \left\{ \mathcal{E}(\mathbf{Z}) := -\sum_{p=1}^{N} \sum_{l=1}^{L} z_{p,l} k(\mathbf{x}_p, \mathbf{m}_l) + \frac{\lambda}{2} \sum_{p,q} k(\mathbf{x}_p, \mathbf{x}_q) \|\mathbf{z}_p - \mathbf{z}_q\|^2 \right\}$$

$$\mathbf{m}_l = \arg\max_{\mathbf{y} \in \mathbf{X}} \sum_p z_{p,l} k(\mathbf{x}_p, \mathbf{y})$$

$$\mathbf{1}^t \mathbf{z}_p = 1, \ \mathbf{z}_p \in \{0,1\}^L \ \forall p \tag{1}$$

where, for each point $p$, $\mathbf{z}_p = [z_{p,1}, \ldots, z_{p,L}]^t$ denotes a binary assignment vector, which is constrained to be within the $L$-dimensional simplex: $z_{p,l} = 1$ if $p$ belongs to cluster $l$ and $z_{p,l} = 0$ otherwise. $\mathbf{Z}$ is the $N \times L$ matrix whose rows are given by the $\mathbf{z}_p$'s. $k(\mathbf{x}_p, \mathbf{x}_q)$ are pairwise affinities, which can be either learned or evaluated in an unsupervised way via a kernel function.

Model (1) integrates several powerful and well-known ideas in clustering. First, it identifies density modes [18, 8], as in popular mean-shift. Prototype $\mathbf{m}_l$ is a cluster mode and, therefore, a valid data

point in the input set. This is important for manifold-structured, high-dimensional inputs such as images, where simple parametric prototypes such as the means, as in K-means, may not be good representatives of the data; see Fig. 1. Second, the pairwise term in $\mathcal{E}$ is the well-known graph Laplacian regularizer, which can be equivalently written as $\lambda \text{tr}(\mathbf{Z}^t \mathbf{L} \mathbf{Z})$, with $\mathbf{L}$ the Laplacian matrix corresponding to affinity matrix $\mathbf{K} = [k(\mathbf{x}_p, \mathbf{x}_q)]$. Laplacian regularization encourages nearby data points to have similar latent representations (e.g., assignments) and is widely used in spectral clustering [32, 26] as well as in semi-supervised and/or representation learning [2]. Therefore, the model can handle non-convex (or manifold-structured) clusters, unlike standard prototype-based clustering techniques such as K-means. Finally, the explicit cluster assignments yield straightforward out-of-sample extensions, unlike spectral clustering [3].

Optimization problem (1) is challenging due to the simplex/integer constraints and the non-linear/non-differentiable dependence of modes $\mathbf{m}_l$ on assignment variables. In fact, it is well known that optimizing the pairwise Laplacian term over discrete variables is NP-hard [30], and it is common to relax the integer constraint. For instance, [33] replaces the integer constraint with a probability-simplex constraint, which results in a convex relaxation of the Laplacian term. Unfortunately, such a direct convex relaxation requires solving for $N \times L$ variables all together. Furthermore, it requires additional projections onto the $L$-dimensional simplex, with a quadratic complexity with respect to $L$. Therefore, as we will see in our experiments, the relaxation in [33] does not scale up for large-scale problems (i.e., when $N$ and $L$ are large). Spectral relaxation [32, 27] widely dominates optimization of the Laplacian term subject to balancing constraints in the context of graph clustering[2]. It can be expressed in the form of a generalized Rayleigh quotient, which yields an exact closed-form solution in terms of the $L$ largest eigenvectors of the affinity matrix. It is well-known that spectral relaxation has high computational and memory load for large $N$ as one has to store the $N \times N$ affinity matrix and compute explicitly its eigenvalue decomposition, which has a complexity that is cubic with respect to $N$ for a straightforward implementation and, to our knowledge, super-quadratic for fast implementations [30]. In fact, investigating the scalability of spectral relaxation for large-scale problems is an active research subject [26, 30, 31]. For instance, the studies in [26, 30] investigated deep learning approaches to spectral clustering, so as to ease the scalability issues for large data sets, and the authors of [31] examined the variational Nyström method for large-scale spectral problems, among many other efforts on the subject. In general, computational scalability is attracting significant research interest with the overwhelming widespread of interesting large-scale problems [11]. Such issues are being actively investigated even for the basic K-means algorithm [11, 22].

The K-modes term in (1) is closely related to kernel density based algorithms for mode estimation and clustering, for instance, the very popular mean-shift [8]. The value of $\mathbf{m}_l$ globally optimizing this term for a given fixed cluster $l$ is, clearly, the mode of the kernel density of feature points within the cluster [29]. Therefore, the K-mode term, as in [6, 25], can be viewed as an energy-based formulation of mean-shift algorithms with a fixed number of clusters [29]. Optimizing the K-modes over discrete variable is NP-hard [33], as is the case of other prototype-based models for clustering[3]. One way to tackle the problem is to alternate optimization over assignment variables and updates of the modes, with the latter performed as inner-loop mean-shift iterates, as in [6, 25]. Mean-shift moves an initial random feature point towards the closest mode via gradient ascent iterates, maximizing at convergence the density of feature points. While such a gradient-ascent approach has been very popular for low-dimensional distributions over continuous domains, e.g., image segmentation [8], its use is generally avoided in the context of high-dimensional feature spaces [7]. Mean-shift iterates compute expensive summations over feature points, with a complexity that depends on the dimension of the feature space. Furthermore, the method is not applicable to discrete domains [7] (as it requires gradient-ascent steps), and its convergence is guaranteed only when the kernels satisfy certain conditions; see [8]. Finally, the modes obtained at gradient-ascent convergence are not necessarily valid data points in the input set.

We optimize a tight bound (auxiliary function) of our concave-convex relaxation for discrete problem (1). The bound is the sum of independent functions, each corresponding to a data point $p$. This yields a scalable algorithm for large $N$, which computes independent updates for assignment variables $\mathbf{z}_p$, while guaranteeing convergence to a minimum of the relaxation. Therefore, our bound optimizer can be trivially distributed for large-scale data sets. Furthermore, we show that the density modes can

be obtained as byproducts of assignment variables $\mathbf{z}_p$ via simple maximum-value operations whose additional computational cost is linear in $N$. Our formulation does not need storing a full affinity matrix and computing its eigenvalue decomposition, neither does it perform expensive projection steps and Lagrangian-dual inner iterates for the simplex constraints of each point. Furthermore, unlike mean-shift, our density-mode estimation does not require inner-loop gradient-ascent iterates. It has a complexity independent of feature-space dimension, yields modes that are valid data points in the input set and is applicable to discrete domains and arbitrary kernels. We report comprehensive experiments over various data sets, which show that our algorithm yields very competitive performances in term of optimization quality (i.e., the value of the discrete-variable objective at convergence)[4] and clustering accuracy, while being scalable to large-scale and high-dimensional problems.

## 2 Concave-convex relaxation

We propose the following concave-convex relaxation of the objective in (1):

$$\min_{\mathbf{z}_p \in \nabla_L} \left\{ \mathcal{R}(\mathbf{Z}) := \sum_{p=1}^{N} \mathbf{z}_p^t \log(\mathbf{z}_p) - \sum_{p=1}^{N} \sum_{l=1}^{L} z_{p,l} k(\mathbf{x}_p, \mathbf{m}_l) - \lambda \sum_{p,q} k(\mathbf{x}_p, \mathbf{x}_q) \mathbf{z}_p^t \mathbf{z}_q \right\} \quad (2)$$

where $\nabla_L$ denotes the $L$-dimensional probability simplex $\nabla_L = \{ \mathbf{y} \in [0,1]^L \mid \mathbf{1}^t \mathbf{y} = 1 \}$. It is easy to check that, at the vertices of the simplex, our relaxation in (2) is equivalent to the initial discrete objective in (1). Notice that, for binary assignment variables $\mathbf{z}_p \in \{0,1\}^L$, the first term in (2) vanishes and the last term is equivalent to Laplacian regularization, up to an additive constant:

$$\mathrm{tr}(\mathbf{Z}^t \mathbf{L} \mathbf{Z}) = \sum_p \mathbf{z}_p^t \mathbf{z}_p d_p - \sum_{p,q} k(\mathbf{x}_p, \mathbf{x}_q) \mathbf{z}_p^t \mathbf{z}_q = \sum_p d_p - \sum_{p,q} k(\mathbf{x}_p, \mathbf{x}_q) \mathbf{z}_p^t \mathbf{z}_q, \quad (3)$$

where the last equality is valid only for binary (integer) variables and $d_p = \sum_q k(\mathbf{x}_p, \mathbf{x}_q)$. When we replace the integer constraints $\mathbf{z}_p \in \{0,1\}$ by $\mathbf{z}_p \in [0,1]$, our relaxation becomes different from direct convex relaxations of the Laplacian [33], which optimizes $\mathrm{tr}(\mathbf{Z}^t \mathbf{L} \mathbf{Z})$ subject to probabilistic simplex constraints. In fact, unlike $\mathrm{tr}(\mathbf{Z}^t \mathbf{L} \mathbf{Z})$, which is a convex function[5], our relaxation of the Laplacian term is concave for positive semi-definite (psd) kernels $k$. As we will see later, concavity yields a scalable (parallel) algorithm for large $N$, which computes independent updates for assignment variables $\mathbf{z}_p$. Our updates can be trivially distributed, and do not require storing a full $N \times N$ affinity matrix. These are important computational and memory advantages over direct convex relaxations of the Laplacian [33], which require solving for $N \times L$ variables all together as well as expensive simplex projections, and over common spectral relaxations [32], which require storing a full affinity matrix and computing its eigenvalue decomposition. Furthermore, the first term we introduced in (2) is a convex negative-entropy barrier function, which completely avoids expensive projection steps and Lagrangian-dual inner iterations for the simplex constraints of each point. First, the entropy barrier restricts the domain of each $\mathbf{z}_p$ to non-negative values, which avoids extra dual variables for constraints $\mathbf{z}_p \geq 0$. Second, the presence of such a barrier function yields closed-form updates for the dual variables of constraints $\mathbf{1}^t \mathbf{z}_p = 1$. In fact, entropy-like barriers are commonly used in Bregman-proximal optimization [35], and have well-known computational and memory advantages when dealing with the challenging simplex constraints [35]. Surprisingly, to our knowledge, they are not common in the context of clustering. In machine learning, such entropic barriers appear frequently in the context of conditional random fields (CRFs) [14, 15], but are not motivated from optimization perspective; they result from standard probabilistic and mean-field approximations of CRFs [14].

## 3 Bound optimization

In this section, we derive an iterative bound optimization algorithm that computes independent (parallel) updates of assignment variables $\mathbf{z}_p$ ($\mathbf{z}$-updates) at each iteration, and provably converges to a minimum of relaxation (2). As we will see in our experiments, our bound optimizer yields

consistently lower values of function $\mathcal{E}$ at convergence than the proximal algorithm in [33], while being highly scalable to large-scale and high-dimensional problems. We also show that the density modes can be obtained as byproducts of the $\mathbf{z}$-updates via simple maximum-value operations whose additional computational cost is linear in $N$. Instead of minimizing directly our relaxation $\mathcal{R}$, we iterate the minimization of an auxiliary function, i.e., an upper bound of $\mathcal{R}$, which is tight at the current solution and easier to optimize.

**Definition 1** $\mathcal{A}_i(\mathbf{Z})$ *is an auxiliary function of* $\mathcal{R}(\mathbf{Z})$ *at current solution* $\mathbf{Z}^i$ *if it satisfies:*

$$\mathcal{R}(\mathbf{Z}) \leq \mathcal{A}_i(\mathbf{Z}), \forall \mathbf{Z} \tag{4a}$$

$$\mathcal{R}(\mathbf{Z}^i) = \mathcal{A}_i(\mathbf{Z}^i) \tag{4b}$$

In (4), $i$ denotes the iteration counter. In general, bound optimizers update the current solution $\mathbf{Z}^i$ to the optimum of the auxiliary function: $\mathbf{Z}^{i+1} = \arg\min_{\mathbf{Z}} \mathcal{A}_i(\mathbf{Z})$. This guarantees that the original objective function does not increase at each iteration: $\mathcal{R}(\mathbf{Z}^{i+1}) \leq \mathcal{A}_i(\mathbf{Z}^{i+1}) \leq \mathcal{A}_i(\mathbf{Z}^i) = \mathcal{R}(\mathbf{Z}^i)$. Bound optimizers can be very effective as they transform difficult problems into easier ones [37]. Examples of well-known bound optimizers include the concave-convex procedure (CCCP) [36], expectation maximization (EM) algorithms and submodular-supermodular procedures (SSP) [21], among others. Furthermore, bound optimizers are not restricted to differentiable functions[6], neither do they depend on optimization parameters such as step sizes.

**Proposition 1** *Given current solution* $\mathbf{Z}^i = [z_{p,l}^i]$ *at iteration* $i$, *and the corresponding modes* $\mathbf{m}_l^i = \arg\max_{\mathbf{y} \in \mathbf{X}} \sum_p z_{p,l}^i k(\mathbf{x}_p, \mathbf{y})$, *we have the following auxiliary function (up to an additive constant) for the concave-convex relaxation in* (2) *and psd[7] affinity matrix* $\mathbf{K}$:

$$\mathcal{A}_i(\mathbf{Z}) = \sum_{p=1}^N \mathbf{z}_p^t(\log(\mathbf{z}_p) - \mathbf{a}_p^i - \lambda \mathbf{b}_p^i) \tag{5}$$

*where* $\mathbf{a}_p^i$ *and* $\mathbf{b}_p^i$ *are the following L-dimensional vectors:*

$$\mathbf{a}_p^i = [a_{p,1}^i, \ldots, a_{p,L}^i]^t, \text{ with } a_{p,l}^i = k(\mathbf{x}_p, \mathbf{m}_l^i) \tag{6a}$$

$$\mathbf{b}_p^i = [b_{p,1}^i, \ldots, b_{p,L}^i]^t, \text{ with } b_{p,l}^i = \sum_q k(\mathbf{x}_p, \mathbf{x}_q) z_{q,l}^i \tag{6b}$$

**Proof 1** *See supplemental material.*

Notice that the bound in Eq. (5) is the sum of independent functions, each corresponding to a point $p$. Therefore, both the bound and simplex constraints $\mathbf{z}_p \in \nabla_L$ are separable over assignment variables $\mathbf{z}_p$. We can minimize the auxiliary function by minimizing independently each term in the sum over $\mathbf{z}_p$, subject to the simplex constraint, while guaranteeing convergence to a local minimum of (2):

$$\min_{\mathbf{z}_p \in \nabla_L} \mathbf{z}_p^t(\log(\mathbf{z}_p) - \mathbf{a}_p^i - \lambda \mathbf{b}_p^i), \forall p \tag{7}$$

Note that, for each $p$, negative entropy $\mathbf{z}_p^t \log \mathbf{z}_p$ restricts $\mathbf{z}_p$ to be non-negative, which removes the need for handling explicitly constraints $\mathbf{z}_p \geq 0$. This term is convex and, therefore, the problem in (7) is convex: The objective is convex (sum of linear and convex functions) and constraint $\mathbf{z}_p \in \nabla_L$ is affine. Therefore, one can minimize this constrained convex problem for each $p$ by solving the Karush-Kuhn-Tucker (KKT) conditions[8]. The KKT conditions yield a closed-form solution for both primal variables $\mathbf{z}_p$ and the dual variables (Lagrange multipliers) corresponding to simplex constraints

$\mathbf{1}^t\mathbf{z}_p = 1$. Each closed-form update, which globally optimizes (7) and is within the simplex, is given by:

$$\mathbf{z}_p^{i+1} = \frac{\exp(\mathbf{a}_p^i + \lambda\mathbf{b}_p^i)}{\mathbf{1}^t \exp(\mathbf{a}_p^i + \lambda\mathbf{b}_p^i)} \; \forall \, p \tag{8}$$

---

**Algorithm 1:** SLK algorithm

---

**Input** : $\mathbf{X}$, Initial centers $\{\mathbf{m}_l^0\}_{l=1}^L$

**Output** : $\mathbf{Z}$ and $\{\mathbf{m}_l\}_{l=1}^L$

1   $\{\mathbf{m}_l\}_{l=1}^L \leftarrow \{\mathbf{m}_l^0\}_{l=1}^L$

2   **repeat**

3     $i \leftarrow 1$ // iteration index

4     $\{\mathbf{m}_l^i\}_{l=1}^L \leftarrow \{\mathbf{m}_l\}_{l=1}^L$

    // z-updates

5     **foreach** $\mathbf{x}_p$ **do**

6       Compute $\mathbf{a}_p^i$ from (6a)

7       $\mathbf{z}_p^i = \frac{\exp\{\mathbf{a}_p^i\}}{\mathbf{1}^t \exp\{\mathbf{a}_p^i\}}$ // Initialize

8     **end**

9     **repeat**

10       Compute $\mathbf{z}_p^{i+1}$ using (6b) and (8)

11       $i \leftarrow i + 1$

12     **until** *convergence*

13     $\mathbf{Z} = [z_{p,l}^{i+1}]$

    // Mode-updates

14     **if** *SLK-MS* **then**

15       update $\mathbf{m}_l$ using (9) until converges

16     **if** *SLK-BO* **then**

17       $\mathbf{m}_l \leftarrow \arg\max_{\mathbf{x}_p} [z_{p,l}^{i+1}]$

18   **until** *convergence*

19   **return** $\mathbf{Z}, \{\mathbf{m}_l\}_{l=1}^L$

---

The pseudo-code for our Scalable Laplacian K-modes (SLK) method is provided in Algorithm 1. The complexity of each inner iteration in z-updates is $\mathcal{O}(N\rho L)$, with $\rho$ the neighborhood size for the affinity matrix. Typically, we use sparse matrices ($\rho << N$). Note that the complexity becomes $\mathcal{O}(N^2 L)$ in the case of dense matrices in which all the affinities are non-zero. However, the update of each $\mathbf{z}_p$ can be done independently, which enables parallel implementations.

Our SLK algorithm alternates the following two steps until convergence (i.e. until the modes $\{\mathbf{m}_l\}_{l=1}^L$ do not change): (i) z-*updates*: update cluster assignments using expression (8) with the modes fixed and (ii) *Mode-updates*: update the modes $\{\mathbf{m}_l\}_{l=1}^L$ with the assignment variable $\mathbf{Z}$ fixed; see the next section for further details on mode estimation.

### 3.1   Mode updates

To update the modes, we have two options: modes via mean-shift or as byproducts of the z-updates.

*Modes via mean-shift:* This amounts to updating each mode $\mathbf{m}_l$ by running inner-loop mean-shift iterations until convergence, using the current assignment variables:

$$\mathbf{m}_l^{i+1} = \frac{\sum_p z_{p,l} k(\mathbf{x}_p, \mathbf{m}_l^i)\mathbf{x}_p}{\sum_p z_{p,l} k(\mathbf{x}_p, \mathbf{m}_l^i)} \tag{9}$$

*Modes as byproducts of the z-updates:* We also propose an efficient alternative to mean-shift. Observe the following: For each point $p$, $b_{p,l}^i = \sum_q k(\mathbf{x}_p, \mathbf{x}_q)z_{q,l}^i$ is proportional to the kernel density estimate

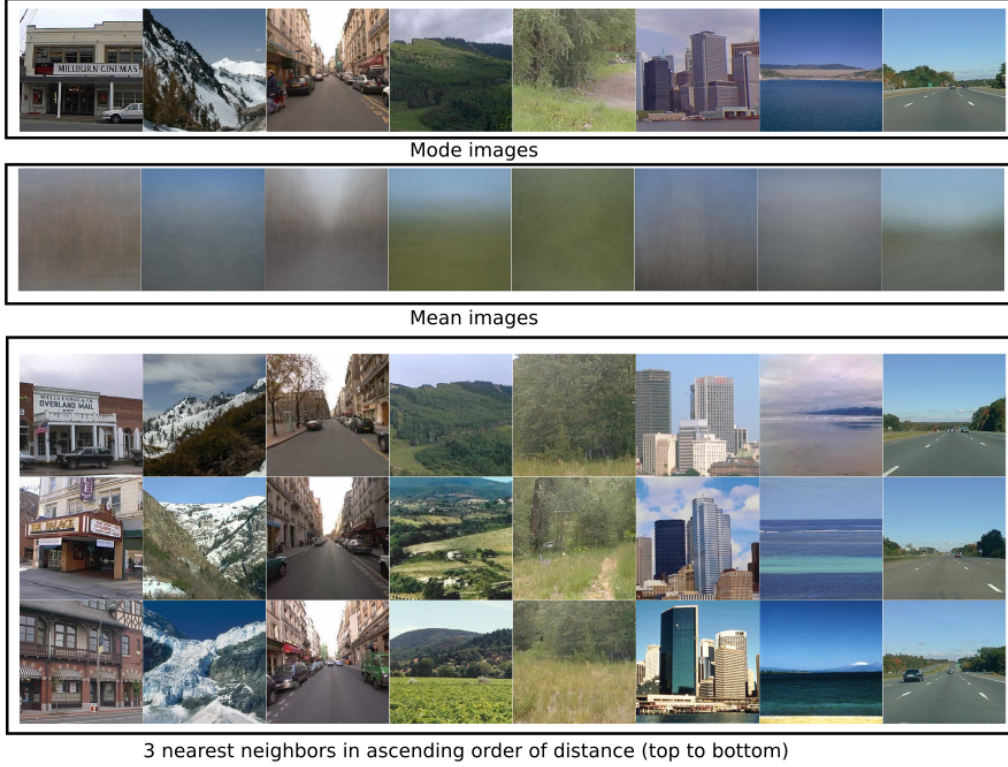

Mode images

Mean images

3 nearest neighbors in ascending order of distance (top to bottom)

Figure 1: Examples of mode images obtained with our SLK-BO, mean images and the corresponding 3-nearest-neighbor images within each cluster. We used LabelMe (Alexnet) dataset.

(KDE) of the distribution of features within current cluster $l$ at point $p$. In fact, the KDE at a feature point $\mathbf{y}$ is:

$$\mathcal{P}_l^i(\mathbf{y}) = \frac{\sum_q k(\mathbf{y}, \mathbf{x}_q) z_{q,l}^i}{\sum_q z_{q,l}^i}.$$

Therefore, $b_{p,l}^i \propto \mathcal{P}_l^i(\mathbf{x}_p)$. As a result, for a given point $p$ within the cluster, the higher $b_{p,l}^i$, the higher the KDE of the cluster at that point. Notice also that $a_{p,l}^i = k(\mathbf{x}_p, \mathbf{m}_l^i)$ measures a proximity between point $\mathbf{x}_p$ and the mode obtained at the previous iteration. Therefore, given the current assignment $\mathbf{z}_p^i$, the modes can be obtained as a proximal optimization, which seeks a high-density data point that does not deviate significantly from the mode obtained at the previous iteration:

$$\max_{\mathbf{y} \in \mathbf{X}} \; [\underbrace{k(\mathbf{y}, \mathbf{m}_l^i)}_{\text{proximity}} + \underbrace{\sum_p z_{p,l} k(\mathbf{x}_p, \mathbf{y})}_{\text{density}}] \qquad (10)$$

Now observe that the $\mathbf{z}$-updates we obtained in Eq. (8) take the form of *softmax* functions. Therefore, they can be used as soft approximations of the hard max operation in Eq. (10):

$$\mathbf{m}_l^{i+1} = \mathbf{x}_p, \; \text{with } p = \arg\max_q [z_{q,l}]^i \qquad (11)$$

This yields modes as byproducts of the $\mathbf{z}$-updates, with a computational cost that is linear in $N$. We refer to the two different versions of our algorithm as SLK-MS, which updates the modes via mean-shift, and SLK-BO, which updates the modes as byproducts of the $\mathbf{z}$-updates.

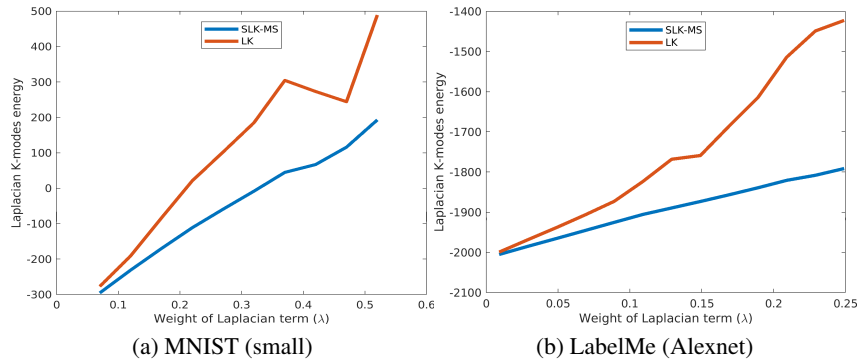

(a) MNIST (small)                    (b) LabelMe (Alexnet)

Figure 2: Discrete-variable objective (1): Comparison of the objectives obtained at convergence for SLK-MS (ours) and LK [33]. The objectives at convergence are plotted versus different values of parameter $\lambda$.

## 4   Experiments

We report comprehensive evaluations of the proposed algorithm[9] as well as comparisons to the following related baseline methods: Laplacian K-modes (LK) [33], K-means, NCUT [27], K-modes [25, 5], Kernel K-means (KK-means) [9, 29] and Spectralnet [26]. Our algorithm is evaluated in terms of performance and optimization quality in various clustering datasets.

Table 1: Datasets used in the experiments.

| Datasets | Samples ($N$) | Dimensions ($D$) | Clusters ($L$) | Imbalance |
|---|---|---|---|---|
| MNIST (small) | $2,000$ | $784$ | $10$ | $1$ |
| MNIST (code) | $70,000$ | $10$ | $10$ | $\sim 1$ |
| MNIST | $70,000$ | $784$ | $10$ | $\sim 1$ |
| MNIST (GAN) | $70,000$ | $256$ | $10$ | $\sim 1$ |
| Shuttle | $58,000$ | $9$ | $7$ | $4,558$ |
| LabelMe (Alexnet) | $2,688$ | $4,096$ | $8$ | $1$ |
| LabelMe (GIST) | $2,688$ | $44,604$ | $8$ | $1$ |
| YTF | $10,036$ | $9,075$ | $40$ | $13$ |
| Reuters (code) | $685,071$ | $10$ | $4$ | $\sim 5$ |

### 4.1   Datasets and evaluation metrics

We used image datasets, except Shuttle and Reuters. The overall summary of the datasets is given in Table 1. For each dataset, imbalance is defined as the ratio of the size of the biggest cluster to the size of the smallest one. We use three versions of MNIST [17]. MNIST contains all the $70,000$ images, whereas MNIST (small) includes only $2,000$ images by randomly sampling $200$ images per class. We used small datasets in order to compare to LK [33], which does not scale up for large datasets. For MNIST (GAN), we train the GAN from [12] on $60,000$ training images and extract the $256$-dimensional features from the discriminator network for the $70,000$ images. The publicly available autoencoder in [13] is used to extract $10$-dimensional features as in [26] for MNIST (code) and Reuters (code). LabelMe [23] consists of $2,688$ images divided into $8$ categories. We used the pre-trained AlexNet [16] and extracted the $4096$-dimensional features from the fully-connected layer. To show the performances on high-dimensional data, we extract $44604$-dimensional GIST features [23] for the LabelMe dataset. Youtube Faces (YTF) [34] consists of videos of faces with $40$ different subjects.

Table 2: Clustering results as NMI/ACC in the upper half and average elapsed time in seconds (s). (*) We report the results of Spectralnet with Euclidean-distance affinity for MNIST (code) and Reuters (code) from [26].

| Algorithm | MNIST | MNIST (code) | MNIST (GAN) | LabelMe (Alexnet) | LabelMe (GIST) | YTF | Shuttle | Reuters |
|---|---|---|---|---|---|---|---|---|
| K-means | 0.53/0.55 | 0.66/0.74 | 0.68/0.75 | 0.81/0.90 | 0.57/0.69 | 0.77/0.58 | 0.22/0.41 | **0.48**/0.73 |
| K-modes | 0.56/0.60 | 0.67/0.75 | 0.69/0.80 | 0.81/0.91 | 0.58/0.68 | 0.79/0.62 | 0.33/0.47 | **0.48**/0.72 |
| NCUT | 0.74/0.61 | 0.84/0.81 | 0.77/0.67 | 0.81/**0.91** | 0.58/0.61 | 0.74/0.54 | 0.47/0.46 | - |
| KK-means | 0.53/0.55 | 0.67/0.80 | 0.69/0.68 | 0.81/0.90 | 0.57/0.63 | 0.71/0.50 | 0.26/0.40 | - |
| LK | - | - | - | 0.81/**0.91** | 0.59/0.61 | 0.77/0.59 | - | - |
| Spectralnet* | - | 0.81/0.80 | - | - | - | - | - | 0.46/0.65 |
| SLK-MS | **0.80**/0.79 | 0.88/0.95 | 0.86/**0.94** | **0.83/0.91** | **0.61/0.72** | **0.82/0.65** | 0.45/0.70 | 0.43/**0.74** |
| SLK-BO | 0.77/**0.80** | **0.89/0.95** | **0.86/0.94** | **0.83/0.91** | **0.61/0.72** | 0.80/0.64 | **0.51/0.71** | 0.43/**0.74** |
| K-means | 119.9s | **16.8**s | 51.6s | 11.2s | 132.1s | 210.1s | 1.8s | 36.1s |
| K-modes | 90.2s | 20.2s | 20.3s | 7.4s | 12.4s | 61.0s | **0.5**s | 51.6s |
| NCUT | 26.4s | 28.2s | **9.3**s | 7.4s | 10.4s | 19.0s | 27.4s | - |
| KK-means | 2580.8s | 1967.9s | 2427.9s | 4.6s | 17.2s | 40.2s | 1177.6s | - |
| LK | - | - | - | 33.4s | 180.9s | 409.0s | - | - |
| Spectralnet* | - | 3600.0s | - | - | - | - | - | 9000.0s |
| SLK-MS | 101.2s | 82.4s | 37.3s | 4.7s | 37.0s | 83.3s | 3.8s | **12.5**s |
| SLK-BO | **14.2**s | 23.1s | 10.3s | **1.8**s | **7.1**s | **12.4**s | 1.3s | 53.1s |

To evaluate the clustering performance, we use two well adopted measures: Normalized Mutual Information (NMI) [28] and Clustering Accuracy (ACC) [26, 10]. The optimal mapping of clustering assignments to the true labels are determined using the Kuhn-Munkres algorithm [20].

## 4.2 Implementation details

We built kNN affinities as follows: $k(\mathbf{x}_p, \mathbf{x}_q) = 1$ if $\mathbf{x}_q \in \mathcal{N}_p^\rho$ and $k(\mathbf{x}_p, \mathbf{x}_q) = 0$ otherwise, where $\mathcal{N}_p^\rho$ is the set of the $\rho$ nearest neighbors of data point $\mathbf{x}_p$. This yields a sparse affinity matrix, which is efficient in terms of memory and computations. In all of the datasets, we fixed $\rho = 5$. For the large datasets such as MNIST, Shuttle and Reuters, we used the *Flann* library [19] with the KD-tree algorithm, which finds approximate nearest neighbors. For the other smaller datasets, we used an efficient implementation of exact nearest-neighbor computations. We used the Euclidean distance for finding the nearest neighbors. We used the same sparse $\mathbf{K}$ for the pairwise-affinity algorithms we compared with, i.e., NCUT, KK-means, Laplacian K-modes. Furthermore, for each of these baseline methods, we evaluated the default setting of affinity construction with tuned $\sigma$, and report the best result found. Mode estimation is based on the Gaussian kernel $k(\mathbf{x}, \mathbf{y}) = e^{-(\|\mathbf{x}-\mathbf{y}\|^2/2\sigma^2)}$, with $\sigma^2$ estimated as: $\sigma^2 = \frac{1}{N\rho} \sum_{\mathbf{x}_p \in \mathbf{X}} \sum_{\mathbf{x}_q \in \mathcal{N}_p^\rho} \|\mathbf{x}_p - \mathbf{x}_q\|^2$. Initial centers $\{\mathbf{m}_l^0\}_{l=1}^L$ are based on K-means++ seeds [1]. We choose the best initial seed and regularization parameter $\lambda$ empirically based on the accuracy over a validation set (10% of the total data). The $\lambda$ is determined from tuning in a small range from $1$ to $4$. In SLK-BO, we take the starting mode $\mathbf{m}_l$ for each cluster from the initial assignments by simply following the mode definition in (1). In Algorithm 1, all assignment variables $\mathbf{z}_p$ are updated in parallel. We run the publicly released codes for K-means [24], NCUT [27], Laplacian K-modes [4], Kernel K-means[10] and Spectralnet [26].

## 4.3 Clustering results

Table 2 reports the clustering results, showing that, in most of the cases, our algorithms SLK-MS and SLK-BO yielded the best NMI and ACC values. For MNIST with the raw intensities as features, the proposed SLK achieved almost 80% NMI and ACC. With better learned features for MNIST (code) and MNIST (GAN), the accuracy (ACC) increases up to 95%. For the MNIST (code) and Reuters (code) datasets, we used the same features and Euclidean distance based affinity as Spectralnet [26], and obtained better NMI/ACC performances. The Shuttle dataset is quite imbalanced and, therefore,

Table 3: Discrete-variable objectives at convergence for LK [33] and SLK-MS (ours).

| Datasets | LK [33] | SLK-MS (ours) |
|---|---|---|
| MNIST (small) | 273.25 | 67.09 |
| LabelMe (Alexnet) | $-1.513\,84 \times 10^3$ | $-1.807\,77 \times 10^3$ |
| LabelMe (GIST) | $-1.954\,90 \times 10^3$ | $-2.024\,10 \times 10^3$ |
| YTF | $-1.000\,32 \times 10^4$ | $-1.000\,35 \times 10^4$ |

all the baseline clustering methods fail to achieve high accuracy. Notice that, in regard to ACC for the Shuttle dataset, we outperformed all the methods by a large margin.

One advantage of our SLK-BO over standard prototype-based models is that the modes are valid data points in the input set. This is important for manifold-structured, high-dimensional inputs such as images, where simple parametric prototypes such as the means, as in K-means, may not be good representatives of the data; see Fig. 1.

## 4.4 Comparison in terms of optimization quality

To assess the optimization quality of our optimizer, we computed the values of discrete-variable objective $\mathcal{E}$ in model (1) at convergence for our concave-convex relaxation (SLK-MS) as well as for the convex relaxation in [33] (LK). We compare the discrete-variable objectives for different values of $\lambda$. For a fair comparison, we use the same initialization, $\sigma$, $k(\mathbf{x}_p, \mathbf{x}_q)$, $\lambda$ and mean-shift modes for both methods. As shown in the plots in Figure 2, our relaxation consistently obtained lower values of discrete-variable objective $\mathcal{E}$ at convergence than the convex relaxation in [33]. Also, Table 3 reports the discrete-variable objectives at convergence for LK [33] and SLK-MS (ours). These experiments suggest that our relaxation in Eq. (2) is tighter than the convex relaxation in [33]. In fact, Eq. (3) also suggests that our relaxation of the Laplacian term is tighter than a direct convex relaxation (the expression in the middle in Eq. (3)) as the variables in term $\sum_p d_p \mathbf{z}_p^t \mathbf{z}_p$ are not relaxed in our case.

## 4.5 Running Time

The running times are given at the bottom half of Table 2. All the experiments (our methods and the baselines) were conducted on a machine with Xeon E5-2620 CPU and a Titan X Pascal GPU. We restrict the multiprocessing to at most 5 processes. We run each algorithm over 10 trials and report the average running time. For high-dimensional datasets, such as LabelMe (GIST) and YTF, our method is much faster than the other methods we compared to. It is also interesting to see that, for high dimensions, SLK-BO is faster than SLK-MS which uses mean-shift for mode estimation.

## 5 Conclusion

We presented Scalable Laplacian K-modes (SLK), a method for joint clustering and density mode estimation, which scales up to high-dimensional and large-scale problems. We formulated a concave-convex relaxation of the discrete-variable objective, and solved the relaxation with an iterative bound optimization. Our solver results in independent updates for cluster-assignment variables, with guaranteed convergence, thereby enabling distributed implementations for large-scale data sets. Furthermore, we showed that the density modes can be estimated directly from the assignment variables using simple maximum-value operations, with an additional computational cost that is linear in the number of data points. Our solution removes the need for storing a full affinity matrix and computing its eigenvalue decomposition. Unlike the convex relaxation in [33], it does not require expensive projection steps and Lagrangian-dual inner iterates for the simplex constraints of each point. Furthermore, unlike mean-shift, our density-mode estimation does not require inner-loop gradient-ascent iterates. It has a complexity independent of feature-space dimension, yields modes that are valid data points in the input set and is applicable to discrete domains as well as arbitrary kernels. We showed competitive performances of the proposed solution in term of optimization quality and accuracy. It will be interesting to investigate joint feature learning and SLK clustering.

## Footnotes

*Corresponding author email: `imtiaz-masud.ziko.1@etsmtl.ca`

[2]Note that spectral relaxation is not directly applicable to the objective in (1) because of the presence of the K-mode term.

[3]In fact, even the basic K-means problem is NP-hard.

[4]We obtained consistently lower values of function $\mathcal{E}$ at convergence than the convex-relaxation proximal algorithm in [33].

[5]For relaxed variables, $\mathrm{tr}(\mathbf{Z}^t \mathbf{L} \mathbf{Z})$ is a convex function because the Laplacian is always positive semi-definite.

[6]Our objective is not differentiable with respect to the modes as each of these is defined as the maximum of a function of the assignment variables.

[7]We can consider $\mathbf{K}$ to be psd without loss of generality. When $\mathbf{K}$ is not psd, we can use a diagonal shift for the affinity matrix, i.e., we replace $\mathbf{K}$ by $\tilde{\mathbf{K}} = \mathbf{K} + \delta \mathbf{I}_N$. Clearly, $\tilde{\mathbf{K}}$ is psd for sufficiently large $\delta$. For integer variables, this change does not alter the structure of the minimum of discrete function $\mathcal{E}$.

[8]Note that strong duality holds since the objectives are convex and the simplex constraints are affine. This means that the solutions of the (KKT) conditions minimize the auxiliary function.

[9]Code is available at: https://github.com/imtiazziko/SLK

[10]https://gist.github.com/mblondel/6230787

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
