[Supplementary Material]

# Scalable Laplacian K-modes

## Supplementary Material

**Imtiaz Masud Ziko** *
ÉTS Montreal

**Eric Granger**
ÉTS Montreal

**Ismail Ben Ayed**
ÉTS Montreal

## 1 Proof of Proposition 1

In this supplemental material, we give a detailed proof of proposition 1 in the paper. Recall that our concave-convex relaxation of the discrete Laplacian K-modes objective is:

$$\mathcal{R}(\mathbf{Z}) = \sum_{p=1}^{N} \mathbf{z}_p^t \log(\mathbf{z}_p) - \sum_{p=1}^{N} \sum_{l=1}^{L} z_{p,l} k(\mathbf{x}_p, \mathbf{m}_l) - \lambda \sum_{p,q} k(\mathbf{x}_p, \mathbf{x}_q) \mathbf{z}_p^t \mathbf{z}_q \tag{1}$$

The proposition states that, given current solution $\mathbf{Z}^i = [z_{p,l}^i]$ at iteration $i$, and the corresponding modes $\mathbf{m}_l^i = \arg\max_{\mathbf{y}} \sum_p z_{p,l}^i k(\mathbf{x}_p, \mathbf{y})$, we have the following auxiliary function (up to an additive constant) for concave-convex relaxation (1) and psd affinity matrix $\mathbf{K}$:

$$\mathcal{A}_i(\mathbf{Z}) = \sum_{p=1}^{N} \mathbf{z}_p^t (\log(\mathbf{z}_p) - \mathbf{a}_p^i - \lambda \mathbf{b}_p^i) \tag{2}$$

where $\mathbf{a}_p^i$ and $\mathbf{b}_p^i$ are the following $L$-dimensional vectors:

$$\mathbf{a}_p^i = [a_{p,1}^i, \ldots, a_{p,L}^i]^t, \text{ with } a_{p,l}^i = k(\mathbf{x}_p, \mathbf{m}_l^i) \tag{3a}$$

$$\mathbf{b}_p^i = [b_{p,1}^i, \ldots, b_{p,L}^i]^t, \text{ with } b_{p,l}^i = \sum_q k(\mathbf{x}_p, \mathbf{x}_q) z_{q,l}^i \tag{3b}$$

*Proof:*

Instead of $N \times L$ matrix $\mathbf{Z}$, let us represent our assignment variables with a vector $\mathbf{z} \in [0, 1]^{LN}$, which is of length $L$ multiplied by $N$ and takes the form $[\mathbf{z}_1, \mathbf{z}_2, \ldots, \mathbf{z}_N]$. As in the paper, each $\mathbf{z}_p$ is a vector of dimension $L$ containing the probability variables of all labels for point $p$: $\mathbf{z}_p = [z_{p,1}, \ldots, z_{p,L}]^t$.

Let $\Psi = -\mathbf{K} \otimes \mathbf{I}_N$, where $\otimes$ denotes the Kronecker product and $\mathbf{I}_N$ the $N \times N$ identity matrix. Now, observe that we can write the relaxed Laplacian term in (1) in the following convenient form:

$$-\lambda \sum_{p,q} k(\mathbf{x}_p, \mathbf{x}_q) \mathbf{z}_p^t \mathbf{z}_q = \lambda \mathbf{z}^t \Psi \mathbf{z} \tag{4}$$

Notice that Kronecker product $\Psi$ is negative semi-definite when $\mathbf{K}$ is positive semi-definite. In this case, function $\mathbf{z}^T \psi \mathbf{z}$ is concave and, therefore, is upper bounded by its first-order approximation at current solution $\mathbf{z}^i$ (iteration $i$). In fact, concavity arguments are standard in deriving auxiliary functions for bound-optimization algorithms [1]. With this condition, we have the following auxiliary function for the Laplacian-term relaxation in (1):

$$-\sum_{p,q} k(\mathbf{x}_p, \mathbf{x}_q) \mathbf{z}_p^t \mathbf{z}_q \le (\mathbf{z}^i)^t \Psi \mathbf{z}^i + (\Psi \mathbf{z}^i)^t (\mathbf{z} - \mathbf{z}^i) \tag{5}$$

(a) SLK-MS for MNIST (GAN)

(b) SLK-BO for MNIST (GAN)

(c) SLK-MS for LabelMe (Alexnet)

(d) SLK-BO for LabelMe (Alexnet)

Figure 1: Relaxed SLK objective (2) in the paper: Convergence of the inner iterations of $\mathbf{z}$-updates shown for MNIST (GAN) and LabelMe (Alexnet) datasets.

Now, notice that, for each cluster $l$, the mode is by definition: $\mathbf{m}_l = \arg\max_{\mathbf{y} \in \mathbf{X}} \sum_p z_{p,l} k(\mathbf{x}_p, \mathbf{y})$. Therefore, $\forall \mathbf{y} \in \mathbf{X}$, we have $-\sum_{p=1}^N z_{p,l} k(\mathbf{x}_p, \mathbf{m}_l) \leq -\sum_{p=1}^N z_{p,l} k(\mathbf{x}_p, \mathbf{y})$. Applying this result to $\mathbf{y} = \mathbf{m}_l^i$, we obtain the following auxiliary function on the K-mode term :

$$-\sum_{p=1}^N z_{p,l} k(\mathbf{x}_p, \mathbf{m}_l) \leq -\sum_{p=1}^N z_{p,l} k(\mathbf{x}_p, \mathbf{m}_l^i) \tag{6}$$

Combining (5) and (6), it is easy to see that (2) is an upper bound on our concave-convex relaxation in (1), up to an additive constant[2]. It easy to check that both bounds in (5) and (6) are tight at the current solution. This complete the proof that (2) is an auxiliary function for our concave-convex relaxation, up to an additive constant.

## 2 Convergence of SLK

Figures 1 and 2 show the convergence of the inner and outer iterations of SLK-BO and SLK-MS using MNIST (GAN) and LabelME (Alexnet) datasets. In Figure 1, the relaxed objective (1), i.e., objective (2) in the paper, decreases monotonically and converges within 50/10 iterations of the

[2]The additive constant depends only on the $\mathbf{z}^i$'s, the assignment variables computed at the previous iteration. This additive constant is ignored in the expression of the auxiliary function in Eq. (2).

(a) SLK-MS for MNIST (GAN)    (b) SLK-BO for MNIST (GAN)

(c) SLK-MS for LabelMe (Alexnet)    (d) SLK-BO for LabelMe (Alexnet)

Figure 2: Convergence of the outer iterations (mode updates): For each cluster, the convergence of the outer loop is shown as the difference in mode values within two consecutive outer iterations. The plots are for MNIST (GAN) and LabelMe (Alexnet) datasets.

(**z**-updates) of SLK for MNIST (GAN)/LabelMe (Alexnet). For each cluster, the convergence of the outer loop (mode updates) is shown as the difference in mode values within two consecutive outer iterations. Notice that both SLK-BO and SLK-MS converge within less than 5 outer iterations, with SLK-MS typically taking more outer iterations. This might be due to the fact that SLK-BO updates the modes from valid data points within the input set, whereas SLK-MS updates the modes as local means via mean-shift iterations.

## Footnotes

*Corresponding author email: `imtiaz-masud.ziko.1@etsmtl.ca`

## References

[1] Kenneth Lange, David R Hunter, and Ilsoon Yang. Optimization transfer using surrogate objective functions. *Journal of computational and graphical statistics*, 9(1):1–20, 2000.