[Reviews · NeurIPS 2018]

Reviewer 1



The paper proposes an alternative objective function for the Laplacian k-modes problem (for simultaneous clustering and density estimation). The proposed objective is a bound for the classical objective proposed in the past, which has a certain number of computational advantages. For example, the updates in the iterative algorithm can be done independently for every point and thus the algorithm can be easily parallelized. I found the proposed bound/objective interesting and its computational advantages significant. The experimental results are convincing; it is particularly interesting that the proposed algorithm is producing solutions that are in fact better for the original laplacian k-modes objective that the algorithm designed for that objective.

Reviewer 2



The paper proposed a noval and scalable approach to solve the Laplacian K-modes problem. The primary strategy is to adopt another way to relax the original discrete problem and further minimize a upper bound of the objective. The method is guaranteed to converge and evaluated to show its superiority over some benchmark clustering algorithms in both clustering performance and algorithmic efficiency. The statements are clear and content is easy to follow. The technical part is also correct. My main criticism is about the significance. -First, I am not sure if Laplacian K-modes energy is a widely used metric for clustering. Laplacian K-modes energy follows a form that simultaneously maximizes weights within each cluster (mode part) and minimizes weights across clusters (Lap part). There are several objectives falling in this category like correlation clustering. However, in experiments, the authors only compare LK objective with some very preliminary objectives NCUT, k-means, k-modes that are in other categories of clustering algorithms (NCUT: balanced min-cut; k-means, k-modes: max weights within clusters). - Second, although the proposed approach has good scalability, the derivation is kind of ad-hoc and all tricks are for scalability. In the first-step relaxation, an entropy term is subjectively added to handle the constrains. With this term, the objective can hardly be a tight relaxation of Laplacian K-modes energy. The proposed approach only holds convergence guarantee while do not has performance guarantee and the trick to obtain such convergence is not difficult to think of. Also, the efficient method to estimate mode is also ad-hoc. I think the proposed method may be useful in some practical scenarios but it is not significant enough for a strong acceptance of NIPS. Hence, I give a borderline overall score.

Reviewer 3



This paper proposed a concave-convex relaxation of Laplacian K-modes problem and solves it by optimizing an upper bound of it. The optimization is done in a fashion similar to K-means algorithm: alternating between modes and assignment vectors. Experimental results show that the proposed approach outperforms state-of-the-art performance with shorter running time. Strengths: This paper is well written and easy to read. The proposed relaxation is empirically proved to be tighter than state-of-the-art and its upper bound is easier to solve in parallel and therefore more scalable. Weaknesses: 1. A interesting baseline to compare with (both in terms of running time and iteration complexity) would be some optimizer that optimizes eq (2) directly, which would provide more intuition about the effect of using this upper bound. Response to Authors' Feedback: I totally agree with making an approximation on Eq (2) and I didn't mean to optimize Eq (2) directly since it would be impractical. What I suggested is to take a small problem and compare direct optimizer on Eq (2) and the proposed optimizer, which shows the quality of approximation on Eq (2). 2. Although some baselines wouldn't work on larger datasets, it is still helpful to consider datasets with larger number of clusters, e.g. imagenet with reduced number of samples and feature dimensions. This makes the story more complete.